# Protein Citrullination by Peptidyl Arginine Deiminase/Arginine Deiminase Homologs in Members of the Human Microbiota and Its Recognition by Anti-Citrullinated Protein Antibodies

**DOI:** 10.3390/ijms25105192

**Published:** 2024-05-10

**Authors:** María-Elena Pérez-Pérez, Enrique Nieto-Torres, Juan-José Bollain-y-Goytia, Lucía Delgadillo-Ruíz

**Affiliations:** 1PhD in Basic Science with Biological Orientation, Academic Unit of Biological Sciences, Universidad Autónoma de Zacatecas, Zacatecas 98066, Mexico; perezperez@uaz.edu.mx (M.-E.P.-P.); luciadelgadillo@uaz.edu.mx (L.D.-R.); 2Department of Immunology and Molecular Biology, Academic Unit of Biological Sciences, Universidad Autónoma de Zacatecas, Guadalupe, Zacatecas 98615, Mexico; 3Academic Unit of Human Medicine and Health Sciences, Universidad Autónoma de Zacatecas, Zacatecas 98160, Mexico; queiq_t@hotmail.com

**Keywords:** microbiota, autoimmunity, rheumatoid arthritis, citrullinome, ACPAs, PADs, microbiome

## Abstract

The human microbiome exists throughout the body, and it is essential for maintaining various physiological processes, including immunity, and dysbiotic events, which are associated with autoimmunity. Peptidylarginine deiminase (PAD) enzymes can citrullinate self-proteins related to rheumatoid arthritis (RA) that induce the production of anti-citrullinated protein antibodies (ACPAs) and lead to inflammation and joint damage. The present investigation was carried out to demonstrate the expression of homologs of PADs or arginine deiminases (ADs) and citrullinated proteins in members of the human microbiota. To achieve the objective, we used 17 microbial strains and specific polyclonal antibodies (pAbs) of the synthetic peptide derived from residues 100–200 of human PAD2 (anti-PAD2 pAb), and the recombinant fragment of amino acids 326 and 611 of human PAD4 (anti-PAD4 pAb), a human anti-citrulline pAb, and affinity ACPAs of an RA patient. Western blot (WB), enzyme-linked immunosorbent assay (ELISA), elution, and a test with Griess reagent were used. This is a cross-sectional case–control study on patients diagnosed with RA and control subjects. Inferential statistics were applied using the non-parametric Kruskal–Wallis test and Mann–Whitney U test generated in the SPSS program. Some members of phyla Firmicutes and Proteobacteria harbor homologs of PADs/ADs and citrullinated antigens that are reactive to the ACPAs of RA patients. Microbial citrullinome and homolog enzymes of PADs/ADs are extensive in the human microbiome and are involved in the production of ACPAs. Our findings suggest a molecular link between microorganisms of a dysbiotic microbiota and RA pathogenesis.

## 1. Introduction

The human microbiome corresponds to a set of microorganisms and microbial genomes located in various anatomical regions, and it also includes bacteria, viruses, fungi, and a certain population of archaea [1]. The biomass of the human microbiome is generated by more than 10^14^ bacterial cells located mainly in the intestine [2,3]. A key event for microbial colonization is birth and breastfeeding, after which the fate of microbial components varies during each stage of life [4]. The kinetics in the intestinal microbial ecosystem is complex since it depends on the diversity, abundance, and temporality of species, and its physiological effects depend mainly on their transcriptional and metabolic potential [5]. In healthy people, approximately 70% of the species are beneficial and have a symbiotic relationship with the host, but they decrease considerably depending on health status [6,7]. Subdominant microbiota belong to transient species with high mobility, mainly bacteria that may have potential pathogenicity and cause harm to the host [8]. This complex relationship can be altered by numerous variables, including infections, nutrition, exposure to antibiotics, and a number of comorbidities that alter the balance of microorganisms, causing changes in activity and diversity, a condition known as dysbiosis [9,10]. The dominant microbial phyla are Firmicutes, Bacteroidetes, Actinobacteria, and Proteobacteria [11]. Other phyla that are found in smaller numbers are Verrucomicrobia, Fusobacteria [12], and Ascomycota, whose greater fungal richness has been found in women and has been associated with inflammatory diseases [13]. Alterations in the phyla Firmicutes, Bacteroidetes, Actinobacteria, and Proteobacteria have also been documented to lead to various autoimmune diseases (AIDs) [11].

### 1.1. Influence of the Microbiome on Autoimmunity

The cause of AIDs is due to a failure in immunological tolerance that triggers an immune response to the molecules themselves [14]. Several studies have revealed that intestinal microbiota may be the origin of the breakdown of tolerance, as well as contribute to the development of these disorders [15]. The immune system and the microbiota collaborate to integrate the innate and adaptive arms of immunity so that immunological and tolerogenic responses are determined and terminated in the best way [16]. It has been proposed that when this mutualistic relationship is impaired by dysbiotic events, dysregulated local and systemic immune responses occur [17,18]. The mechanisms that connect intestinal dysbiosis with autoimmune pathways involve: Dysfunction of intestinal dendritic cells [19], innate lymphoid cells [20], and mechanisms related to phagocytic function [21], alteration in intestinal mast cell density, C macrophage activation [22], damage to the intestinal epithelial barrier, imbalance of CD4+ cells, depletion of CD8+ cells, alteration in the marginal B-cell zone, dysfunction of plasmacytoid DCs, and invariant natural killer cells [23]. The decrease in butyrate derived from the microbiota reduces cellular metabolism, affects the memory of activated CD8 ^+^ T cells, and impairs optimal recovery responses after antigen re-encounter [24].

Other mechanisms linking dysbiosis with autoimmunity are microbial translocation and molecular mimicry that causes a cross-activation of immunomodulatory cells to autoantigens [17]. However, the genetics of the host are crucial for the detection of antigens related to human leukocyte antigens (HLAs); they are essential clues for triggering autoimmune responses [25]. Sometimes, the tissue target is distant from the intestine, as is the case of RA, which affects 1% of the general population and is characterized by the presence of T and B autoreactive cells against autoantigens mainly present in synovial tissue. The participation of HLA class II alleles that encode the alpha and beta protein chains expressed on the cell surface of antigen-presenting cells (APC) is crucial in RA pathogenesis. The DRB1*0401 allele, which encodes a particular sequence located between residues 60–74 of hypervariable region 3 of the beta chain, is known as a “shared epitope,” and this domain is critical for the binding of autoantigens, particularly those with citrullinated residues. Therefore, the “shared epitope” appears to increase the susceptibility of individuals to develop RA, since 50% of patients hosting this allele are positive for rheumatoid factor (RF) and 90% develop severe rheumatoid complications or extra-articular manifestations [26]. The shared epitope allows the translocation of calreticulin, which in turn increases calcium-dependent signaling that activates the PAD pathway [27].

Mammalian PADs belong to a peculiar family of enzymes that remove peptidylarginine from peptidyl-citrulline residues in the process of protein citrullination. There are five isoforms of human PADs, but PAD2 and PAD4 have been implicated in RA due to their presence in immune cells and synovial tissue [28,29]. PAD4 citrullinate proteins through neutrophil extracellular traps (NETs), which are important for antimicrobial responses, but also increase in RA, boosting autoimmune responses [30,31,32]. PAD2 is not required for NET formation but is required for joint citrullination [33]. Recently, it has been suggested that macrophage extracellular traps (METs) contribute to the set of citrullinated antigens dependent on PAD2 and PAD4 [34]. The various citrullinated peptides stimulate autoreactive T lymphocytes, which in turn stimulate autoreactive B cells to produce ACPAs. These are one of the biochemical markers used for the diagnosis of RA, with a sensitivity >80% and a specificity of 98% in patients [35]; together, these mechanisms can trigger or contribute to pathogenic changes in synovium and bone [36].

The effects of PAD activity have been explored using high-resolution mass spectrometry, which facilitates the accurate detection of residues’ citrullination, and with the current omics data, it has been possible to provide the expression patterns of PAD isoforms, including their substrates [37]. It has been documented that RA can originate from mucosal locations, including the intestine and oral cavity [38]. Several molecular studies have shown that dysbiosis in patients with RA is characterized by the loss of Bacteroidetes with an overgrowth of Prevotella copri (*P. copri*) in the intestine, also in patients treated with methotrexate [39,40]. This phenomenon is especially common in patients carrying the “shared epitope” [41]. Likewise, it has been shown that filamin A and N-acetylglucosamine-6-sulfatase are identical to epitopes generated by Prevotella [42]. It has also been suggested that infectious microorganisms could be associated with RA, as some patients experience an improvement when treated with antibiotics [43]. Another piece of information that supports the previous assertion is the association of Porphyromonas gingivalis (*P. gingivalis*) that has been widely clinically and experimentally documented. *P. gingivalis* is important in rheumatology due to the enormous capacity of its PAD homolog enzymes for the post-translational modification of potentially arthritogenic peptides [44].

### 1.2. Homologs of PADs/ADs in the Human Microbiome and Their Implications in RA

Studies on the homologs of PAD/AD are limited. At the species level, the only bacterial PAD enzyme described is in *P. gingivalis* (called PPAD) [45]. Subsequently, an evolutionary analysis revealed the phylogenetic conservation of human PAD2 and PAD3 with the AD enzymes from *E. coli* (GenBank: EDV68547.1) and *S. aureus* (GenBank: BBA 25170.1) [46]. ADs are conserved in a variety of bacteria, and the AD pathway provides energy in anaerobic conditions. L-arginine, an essential molecule in the formation of biofilms, is catalyzed by ADs and converted into L-citrulline; subsequently, metabolic conversion steps follow, which finish with the production of ATP, ammonia, and CO_2_ [47]. A recent database analysis identified homologs of PAD/AD in the *phyla Cyanobacteria*, *Actinobacteria*, and *Proteobacteria* [48]. However, the experimental evidence is limited to a few species, and only PPAD has been associated with clinical implications in RA, which has a correlation with greater arthritic severity in mice and greater T-cell activation and expression of cytokines [49,50,51,52,53,54,55,56], citrullinated proteins that are correlated with bone loss, osteoclast precursors [52], cartilage destruction, and inflammatory responses [53]. The generation of ACPAs and IgG responses to specific citrullinated peptides have also been demonstrated with PPADs; it has even been suggested that they could be biomarkers of RA [54,55]. 

Due to the limited experimental evidence on homologs of PAD/AD and microbial citrullinomes, it is feasible to establish the hypothesis that various members of the human microbiota possess the catalytic machinery to citrullinate their proteins to cause the generation of microbial ACPAs and contribute to the breakdown of immunological tolerance in people predisposed to RA. The objective of this research is to study the expression of homologs of PAD/AD and microbial citrullinomes in 17 members of the human microbiota belonging to the phyla Proteobacteria, Firmicutes, and Ascomycota.

## 2. Results

### 2.1. Homologs of PADs/ADs Are Present in Microbial Extracts

Anti-PAD2 and anti-PAD4 pAbs demonstrated the presence of reactive bands in 76% of the microbial extracts tested. A comparison of the molecular masses of the expressed microbial bands revealed that ~35, 45, 75, and 85 kDa were reactive to anti-PAD2 pAb and that ~35 and 45 kDa were reactive to anti-PAD4 pAb. The molecular weights of PADs/ADs exhibit slight differences compared to that of human PAD enzymes. Furthermore, our results suggest that microbial homologs of PAD/AD are equally immunogenic because they are recognized by specific anti-PADs pAbs, probably because some microbial domains share homology or cross-react with mammalian PAD2 and PAD4 enzymes (Figure 1).

### 2.2. Microbial Proteins Are Endogenously Citrullinated

We demonstrated that commercial anti-citrulline pAb recognizes multiple microbial bands. Reactive bands were observed at molecular levels of ~30, 35, 40, 45, and 65 kDa. This result suggests that some microbial peptides are endogenously citrullinated (Figure 2, Figure 3 and Figure 4). 

### 2.3. Isotype IgG of RA Sera Recognizes Antigens of the Microbiota

The RA serum samples were positive for IgG anti-CCP antibodies in the determination by ELISA, with a mean of 390 ± 344 units, and the control group was negative. Since host–microbe mutualism is established after birth, the adaptive immune response involves “peripheral immune education” or selection to tolerate the microbiome, and one mechanism involved is the relationship between secretory IgA antibodies and Bacteroides, which are mainly produced in the intestine. Therefore, we investigated the presence of IgG class antibodies in the serum of RA patients as a fingerprint of dysbiotic events. We found that sera-positive anti-CCP of RA patients showed IgG class antibodies directed against various microbial proteins in contrast to control sera samples that showed lower levels. However, the reactivity to nonspecific bands that may only correspond to anti-microbial antibodies. Furthermore, in these assays, irrelevant levels of IgA antibodies against some microbial proteins were measured in RA patients and controls (Figure 5 and Figure 6). The RF cross-reactivity was ruled out by the repeated absorption of ACPAs, using staphylococcus A, as well as commercial latex coated with normal IgG to absorb possible RF activity.

### 2.4. Affinity ACPAs Recognize Citrullinated Antigens of the Microbiota

By eluting antibodies specific for citrullinated peptides, we were able to demonstrate that affinity-purified ACPAs of RA patients recognize multiple microbial bands of ~30, 35, 40, 45, and 65 kDa, which coincide with bands reactive to the commercial anti-citrulline pAb, revealing a 69% coincidence in Proteobacteria, 50% in Firmicutes, and 40% in Ascomycota (Figure 7). This result reinforces that some microbial peptides are endogenously citrullinated and shows that the ACPAs of RA patients have an affinity for a high percentage of these peptides.

### 2.5. Citrullination Activity in Microbial Cultures

Citrullination activity was observed in 57% of the species corresponding to the *phylum Proteobacteria*, 50% in *Firmicutes*, and 50% in *Ascomycota*. The species that present citrullination coincide with 66% with the species that express citrullinated antigens by *WB* (Table 1 and Figure 8).

## 3. Discussion

The results of the present investigation demonstrate the presence of homologs of PAD/AD in members of the phyla Firmicutes and Proteobacteria. Understandably, these homologs differ in molecular size with respect to mammalian PADs, however the homologs are functional and immunogenic. Our results partially coincide with what was reported in a recent analysis that was carried out on databases, where homologs of PAD/AD were identified in the phyla Cyanobacteria, Actinobacteria, and Proteobacteria [48]. Nevertheless, our results suggest that the phylum Firmicutes should be considered as a new microbial phylum hosting homologs of PAD/AD. On the other hand, the phylum Ascomycota should continue to be explored since, despite the statistical analysis that showed negative values, some species qualitatively express PAD/AD homologs, as well as citrullinated proteins and citrullination activity. Probably, the number of species analyzed corresponding to the phylum Ascomycota should be increased. The phylum Proteobacteria is the one with the greatest number of species that express homologs of PAD/AD, citrullinated antigens, and citrullination activity. At the species level, we demonstrated the expression of PAD/AD homologs in seven microorganisms: *P. mirabilis*, *E. coli*, *K. oxytoca*, *M. morgannii*, *A. baumannii* (Proteobacteria), *S. epidermidis*, and *S. aureus* (Firmicutes). Of the species revealed in our research, only two coincide with species reported to express ADs (*E. coli* and *S. aureus*), with phylogenetic conservation in human PADs [46]. For its part, microbial citrullinome has only been identified in *P. gingivalis* [56] and *E. coli*, revealing metabolic proteins related to stress response [46]. Our results demonstrate citrullinomes in seven microorganisms; interestingly, they are pathogenic, which could indicate a molecular role in dysbiosis. 

The proteins identified in our study could correspond to those reported in *E. coli*, highlighting the 23 kDa ribosomal protein 30S/S4rpsD (Uniprot ID: Q1R636), which binds to 16S rRNA. Moreover, for the mannose-specific EIIAB component of the 35 kDa PTS/manX system (Uniprot ID: P69799), an important active carbohydrate transport system, it also binds to 39 kDa glycerol dehydrogenase/gldA (Uniprot ID: P0A9S6) that catalyzes NAD-dependent oxidation. his enzyme allows microorganisms to use glycerol as a carbon source under anaerobic conditions. Lastly, it binds to fumarate hydratase class I, anaerobic/fumB 60 kDa (Uniprot ID: P14407), which catalyzes the reversible hydration of fumarate to (S)-malate; and the 74 kDa threonine-tRNA ligase that is involved in RNA splicing and transcriptional and translational regulation [46]. 

A total of 66% of the species that showed citrullinomes corresponds to the citrullination activity that was determined in the cultures. Interestingly, some species, such as *P. vulgaris*, express citrullinomes that coincide with citrullination activity; however, they may express other homolog enzymes different to PAD2 or PAD4/ADs. Probably, these microorganisms host other citrullinating molecules that use ornithine to produce citrulline, like *Eggerthella* [57]. Other species that caught our attention were *C. freundii* and *M. morgannii*, which expressed citrullinated antigens by a different route than NO synthesis.

Gram-positive and Gram-negative bacteria produce and release outer membrane vesicles (OMVs) and membrane vesicles (MVs) that participate in cellular communication. These vesicles transport different bacterial molecules [58,59], including citrullinated proteins [46]. It has been determined that the citrullination of microbial proteins can also occur after cell lysis [57], so it is likely that during NETosis and METosis, the expression of human PADs and microbial PADs/ADs increases, inducing the citrullination of autoantigens and microbial antigens. This contributes to the excessive production of ACPAs, both human and microbial, which act in synergy so as to break immunologic tolerance in predisposed patients that suffer from RA. Anoux et al. demonstrated the generation of ACPAs induced by PPAD in *P. gingivalis* [54]. These data are not in disagreement with ours, since the affinity ACPAs obtained through elution recognized other microbial citrullinated targets, which suggest that more microorganisms, mainly pathogenic, play a role in RA pathogenesis.

The fact that there is a presence of IgG class antimicrobial antibodies in RA patients, but there not in the controls, led us to ask the following questions. First, are these antibodies traces of a dysbiotic event? If so, we understand that they are not specific rheumatoid markers, since dysbiosis can occur in any disease. Second, what is the role of these antibodies at the host–microbiome interface? It is clear that there is still much to be answered; however, our research, through a superficial study of microbial proteins, suggests that the homologs of PAD/AD and the microbial citrullinome are widely distributed in the human microbiome and not restricted to *P. gingivalis* and *E. coli.* And, even though it is likely that many species have the ability to generate citrullinated peptides, many of these modified peptides may be irrelevant for inducing arthritis. Another possibility is that they do not share mimotopes or that individuals carrying these citrullinated bacterial peptides lack a “shared epitope”. 

Numerous investigations have focused on the hypothesis that intestinal microbes and the leakage of microbial enzymes can influence a greater citrullination of peptides that leads to the production of ACPAs and inflammation through mechanisms of filtration and migration to the joints, provoking immune responses and synovitis [60]. Molecular studies have shown that bacteria can migrate to the periphery and contribute to synovial inflammation [61,62]. More than 200 citrullinated peptides have been described in proteomic studies in the colonic mucosa of subjects with RA, which included targets of ACPAs, but only three were exclusive to this pathology [63]. However, it is not ruled out that some may have a microbial origin. In the future, it would be interesting to purify and sequence the microbial targets revealed in this study and expand our knowledge of the pathogenic role that the human microbiome plays in RA.

## 4. Materials and Methods

### 4.1. Type of Study

This was a cross-sectional case–control study. This research was observational, quantitative, descriptive, and analytical, and followed the principles of the Helsinki Declaration on Human Experimentation.

### 4.2. Population and Sample

A total of 20 individuals were evaluated: 10 diagnosed with RA, according to the American College of Rheumatology/European League Against Rheumatism 2010 classification criteria. They were from the state of Zacatecas and attended public/private outpatient consultations (9 women and 1 man, with a mean age of 42 ± 14.5 years old), and 10 individuals did not have RA (6 women and 4 men, with a mean age of 42 ± 3.55 years old). All individuals included in this study signed informed consent forms approved by the ethics committee. 

### 4.3. Biological Samples

In the RA and control groups, serological samples were obtained by venipuncture for the measurement of RA biomarkers and antimicrobial reactivity. Microbial strains provided by the microbial laboratories of the General Hospital of Fresnillo, Zac., and the Academic Unit of Chemical Sciences, UAZ, and reference strains type ATCC, were also used in this study, which served as antigenic sources to evaluate the expression of homolog enzymes of PAD/AD and microbial citrullinated proteins, and their subsequent reactivity to RA sera and control sera.

### 4.4. Determination of Anti-CCP Antibodies by ELISA

Serological samples from patients with RA and control sera were subjected to an ELISA test for the determination of anti-CCP antibodies using the Euroimmun commercial kit (EA 15,059,601 G). For this, the 1:100 diluted samples were placed in wells containing citrullinated peptides and incubated for 1 h, followed by extensive washes. Conjugated IgG anti-human antibody HRP was added for 30 min; additional washes were performed; and subsequently, the TMB substrate was added for 15 min. Finally, the reaction was stopped with 0.2 M sulfuric acid (H_2_SO_4_). Absorbances were obtained by spectrophotometry at 450 nm, and calculations were performed to obtain concentration units.

### 4.5. Affinity ACPAs Obtained by Elution

The positive anti-CCP samples of RA patients to obtain affinity ACPAs were used; a commercial ELISA plate with citrullinated peptides was used (Euroimmun Kit EA 15,059,601 G) for the incubation of positive anti-CCP samples for 1 h. After extensive washing, the specific antibodies bound to peptides were eluted with 0.2 M glycine (pH 2.6) and neutralized with Tris Base (pH 8.6). The affinity ACPAs were concentrated and combined with microbial antigens in WB assays.

### 4.6. Microbial Cultures Using Selective and Differential Mediums

The microbial antigens were obtained from the following strains classified into 3 phyla: Proteobacteria (Proteus mirabilis (*P. mirabilis*); Proteus vulgaris (*P. vulgaris*); Escherichia coli 25,922 (*E. coli*); Citrobacter freundii (*C. freundii*); Morganella morgannii (*M. morgannii*)]; Pseudomonas aeruginosa (*P. aeruginosa*); and Acinetobacter baumannii (*A. baumannii*)); Firmicutes (Lactobacillus sp; Enterococcus faecalis (*E. faecalis*); Staphylococcus epidermidis (E. epidermidis); and Staphylococcus aureus 25,923 (*S. aureus*)); and Ascomycota (Candida albicans (*C. albicans*); C. glabrata (*C. glabrata*); Candida tropicalis (*C. tropicalis*); and Sacharomyses cerevisiae (*S. cerevisiae*)). First, the microorganisms were cultured in a solid medium; the following media were used: blood agar, MacConkey agar, mannitol salt agar, CHROMagar agar, BIGGY agar, and Man Rogosa Sharpe (MRS) agar. The microbial strains were grown at 37 °C on a blood agar medium. After inoculation, Petri dishes were first placed in an anaerobic chamber and subsequently in an incubator. The identification of the microorganisms was carried out using the bioMérux VITEK^®^ 2 system. The microbial preservation of 16 species was carried out on Mueller–Hinton agar and the Lactobacillus on MRS agar.

### 4.7. Microbial Antigenic Extraction and Quantification

After growth in solid cultures, 500 µL of microbial colonies were collected, washed in PBS, and centrifuged for 10 min at 5000 rpm. The pellets were then resuspended in 1 mL of lysis buffer (1 mL SDS-1, 0.25 M EDTA, 1 mM Tris Base, pH 7.5, and 5 mM PMSF) and sonicated at 60% efficiency in five cycles. Subsequently, the lysates were centrifuged at 10,000 rpm for 10 min at 4 °C, after which the supernatants were saved. In the case of yeasts, the precipitates were collected and macerated in a mortar adding liquid nitrogen until a fine powder was obtained. Subsequently, the powder was resuspended in a lysis buffer as described. Microbial protein quantification was performed by spectrophotometry using the Bradford method at 595 nm. The microbial protein extracts were stored at −70 °C to be used as antigenic sources in the *WB* assays.

### 4.8. SDS-PAGE and Western Blot Analyses

Microbial protein extracts were characterized by 10% SDS-PAGE [64]. The proteins were standardized to a concentration of 2 mg/mL per well. After electrophoresis, the microbial proteins were transferred to nitrocellulose membranes [65] (NITROCEL MEMB, Bio-Rad, BIO1620115, Hercules, CA, USA) and the nonspecific binding of antibodies was blocked with 5% skimmed-milk powder dissolved in PBS. The homolog enzymes of PAD/AD were identified by applying commercial polyclonal antibodies: rabbit IgG anti-PAD2 (PA5-19474; Invitrogen; 1 µg/mL, Waltham, MA, USA) that recognizes the synthetic peptide derived from residues 100–200 of human PAD2 (Uniprot ID#Q9Y2J8, Geneva, Switzerland), and rabbit IgG anti-PAD4 (PA5-22317; Invitrogen; 1:500 diluted) that recognizes a recombinant fragment corresponding to a region in amino acids 326 and 611 of human PAD4 (Uniprot ID#Q9UM07). Rabbit anti-IgG secondary antibody was used at a dilution of 1:10,000, HRP-conjugated (ab97051 from Abcam, Cambridge, UK). The citrullinated proteins in the microbial extracts were detected by a human IgG anti-citrulline commercial polyclonal antibody (ab100932) at a 1:500 dilution rate and a human anti-IgG secondary antibody diluted at 1:10,000, HRP-conjugated (ab 6858 Abcam).

The antimicrobial reactivity and human immunoglobulin isotype involving RA sera and control sera were used at a dilution rate of 1:2000, and anti-IgG human (ab 6858 Abcam) and anti-IgA human (A18781 from Invitrogen) secondary antibodies diluted at 1:10,000 (HRP-conjugated). Additionally, the affinity ACPAs of RA patients were used to evaluate the reactivity against citrullinated microbial antigens labeled by human anti-citrulline pAb. Finally, for the normalization with the constitutive protein, the anti-HSP70 monoclonal antibody from Invitrogen (MA5-45208) was used at a 1:1000 dilution and anti-mouse secondary antibody at a 1:10,000 dilution, HRP-conjugated. The primary antibodies were diluted in 3% milk-PBS for 18 h at 4 °C and were incubated with gentle rocking. Secondary antibodies conjugated with peroxidase were diluted in 3% milk-PBS for 2 h at 4 °C and were incubated in the dark. Immunoreactive bands were detected by chemiluminescence according to the manufacturer’s recommendations (Clarity™ Western ECL Substrate Cat # 170-5061 BIO RAD) on a ChemiDoc XRS Molecular Imager (BIO-RAD). The intensity of bands is represented in pixels.

### 4.9. Citrullination Activity in Microbial Cultures

The citrullination activity of the microorganisms was performed using the Griess Reagent test. The production of L-Citrulline was determined through the synthesis of NO [66]. To achieve this, the microorganisms were cultured in solid medium, as previously described. A total of 1 mL of Griess reagent (087K5013 SIGMA, St. Louis, MO, USA) was added to the Petri dishes corresponding to each species; this was left for 15 min to later produce a volume of 300 µL and then placed in ELISA wells. The color change was interpreted as the production of L-Citrulline, from a yellow or brown hue (depending on the solid medium, MH, or MRS agar) to a pinkish-reddish hue. The absence of a pinkish-reddish color is a sign of the inexistence of L-Citrulline.

### 4.10. Statistic Analysis

Inferential statistics were used to compare the expression of PAD/AD homologs as well as citrullinated antigens in microbial extracts. The data are expressed as median and ranges. When comparing the three groups, the Kruskal–Wallis test was used. The non-parametric Mann–Whitney U test was used to compare the antimicrobial reactivity RA group vs. control group. A *p*-value ≤ 0.05 was considered statistically significance. All tests and graphs were generated using SPSS version 24.

## 5. Conclusions

Some members of the phyla Firmicutes and Proteobacteria host homologs of PADs/ADs and citrullinated antigens are reactive to the ACPAs of RA patients. This suggests that microbial citrullinome and homolog enzymes of PAD/AD are extensive in the human microbiome and are involved in the production of ACPAs. Our findings suggest a molecular link between microorganisms of dysbiotic microbiota and RA pathogenesis.

## Figures and Tables

**Figure 1 ijms-25-05192-f001:**
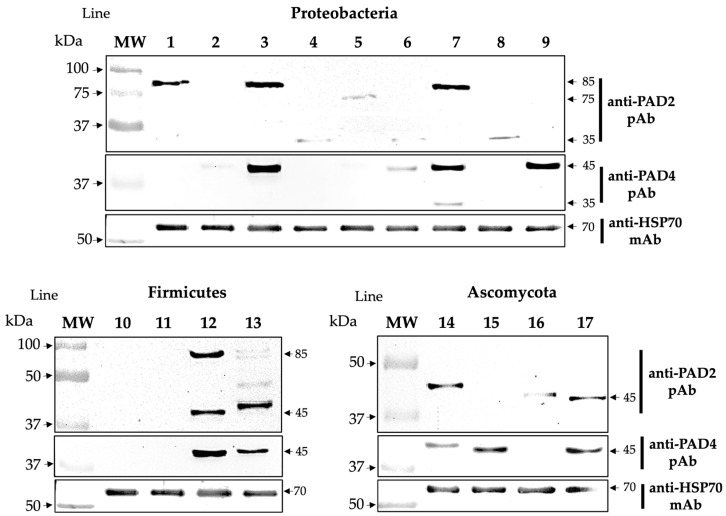
Homologs of PAD/AD present in microbial extracts. The expression of homolog enzymes of PAD/AD shown in species belonging to the phyla Proteobacteria, Firmicutes, and Ascomycota. In the lower part of each blot, there are bands recognized by anti-HSP70 mAb that were used as the constitutive protein control of each microbial extract. Lines correspond to 1. *P. mirabilis*, 2. *P. vulgaris*, 3. *E. coli*, 4. *K. pneumoniae*, 5. *K. oxytoca*, 6. *C. freundii*, 7. *M. morgannii*, 8. *P. aeruginosa*, 9. *A. baumannii*, 10. *Lactobacillus* sp., 11. *E. faecalis*, 12. *S. epidermidis*, 13. *S. aureus*, 14. *C. albicans*, 15. *C. glabrata*, 16. *C. tropicalis*, and 17. *S. cerevisiae*.

**Figure 2 ijms-25-05192-f002:**
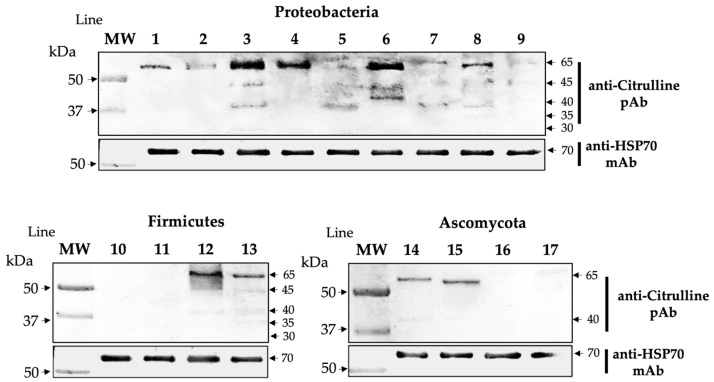
Citrullinated antigens are present in microbial extracts. Expression of citrullinated antigens is shown by commercial anti-citrulline pAb. In the lower part of the panel is the constitutive protein HSP70 expressed in each microbial extract. The lines correspond to proteins from different species: 1. *P. mirabilis*, 2. *P. vulgaris*, 3. *E. coli*, 4. *K. pneumoniae*, 5. *K. oxytoca*, 6. *C. freundii*, 7. *M. morgannii*, 8. *P. aeruginosa*, 9. *A. baumannii*, 10. *Lactobacillus sp*, 11. *E. faecalis*, 12. *S. epidermidis*, 13. *S. aureus*, 14. *C. albicans*, 15. *C. glabrata*, 16. *C. tropicalis*, and 17. *S. cerevisiae*.

**Figure 3 ijms-25-05192-f003:**
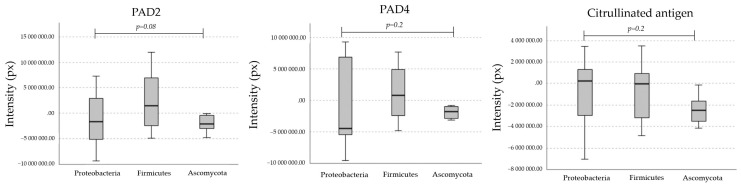
Expression of homologs of PAD2/PAD4 and citrullinated proteins in the phyla Proteobacteria, Firmicutes, and Ascomycota. A value of *p* ≤ 0.05 is considered statistically significant according to the Kruskal–Wallis test. Px = pixels. After normalization with the constitutive protein, the statistical analysis allowed us to differentiate the phyla with positive and negative values.

**Figure 4 ijms-25-05192-f004:**
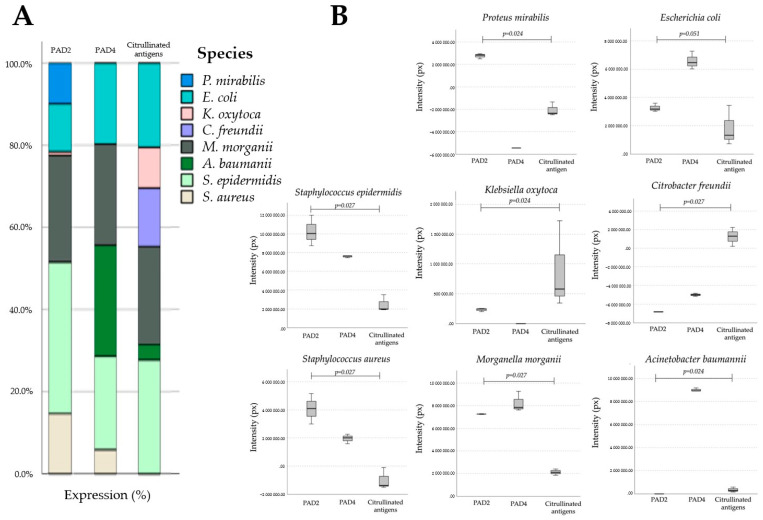
Comparative expression of homologs of PAD2, PAD4, and citrullinated antigens in microbial extracts. The stacked bar graph (**A**) represents the percentage of expression by species, while (**B**) shows the mean pixel intensity of the bands reactive to anti-PAD2 pAb, anti-PAD4 pAb, and anti-citrulline pAb, with values of *p* ≤ 0.05 that are considered significant according to the Kruskal–Wallis test. Px = pixels. The statistical analysis allowed us to represent only the species that had positive expression values after normalization with the constitutive protein HSP70.

**Figure 5 ijms-25-05192-f005:**
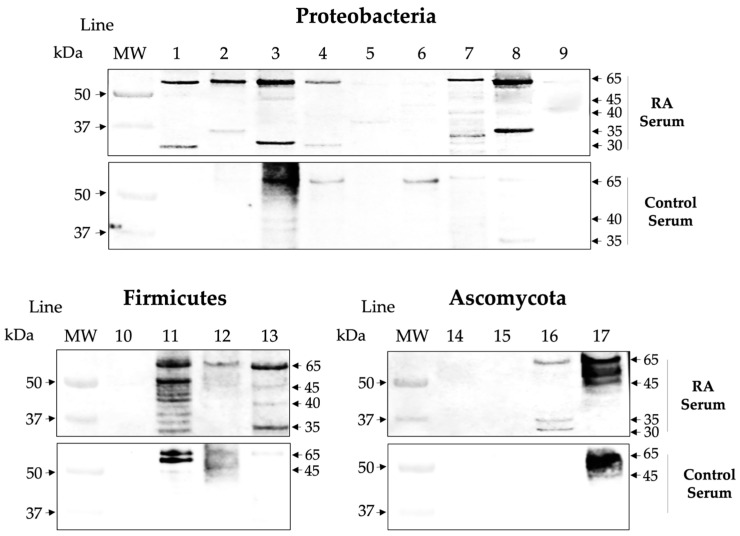
Differences in reactivity to microbial extracts of 17 commensals between RA sera and controls. A representative panel of *WB* shows the immunological recognition of IgG in RA sera and lower reactivity in control sera. The reactivity to nonspecific bands that may only correspond to anti-microbial antibodies. The lines correspond to proteins from different species: 1. *P. mirabilis*, 2. *P. vulgaris*, 3. *E. coli*, 4. *K. pneumoniae*, 5. *K. oxytoca*, 6. *C. freundii*, 7. *M. morgannii*, 8. *P. aeruginosa*, 9. *A. baumannii*, 10. *Lactobacillus* sp., 11. *E. faecalis*, 12. *S. epidermidis*, 13. *S. aureus*, 14. *C. albicans*, 15. *C. glabrata*, 16. *C. tropicalis*, and 17. *S. cerevisiae*.

**Figure 6 ijms-25-05192-f006:**
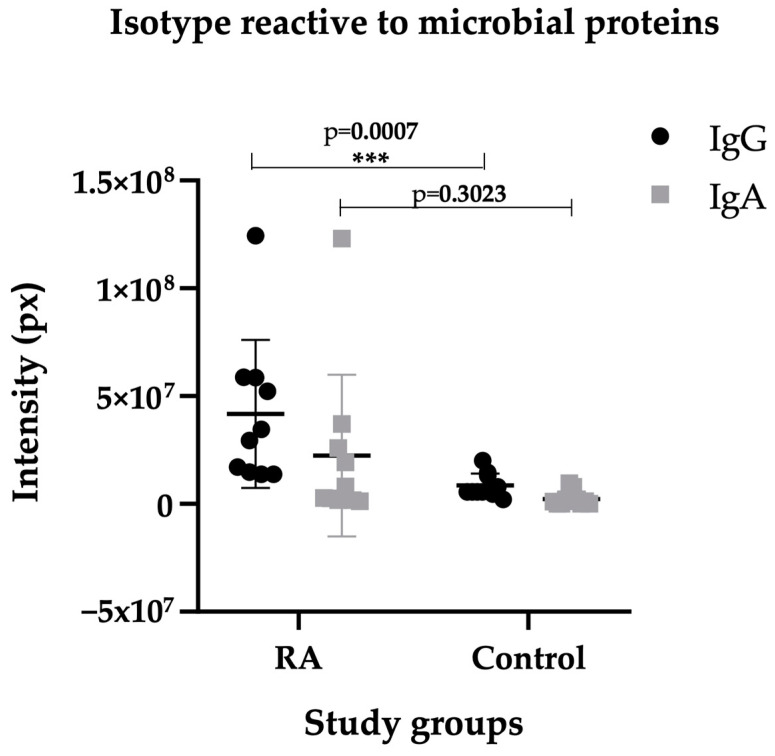
Isotype of immunoglobulins reactive to microbial extracts in RA and control sera. Note that the IgG isotype is mainly involved in the reactivity of microbial proteins, and the difference between the two groups is significant (*p* = 0.0007), while the reactivity of the IgA isotype is (*p* = 0.3023) according to the Mann–Whitney test. *** highly significant.

**Figure 7 ijms-25-05192-f007:**
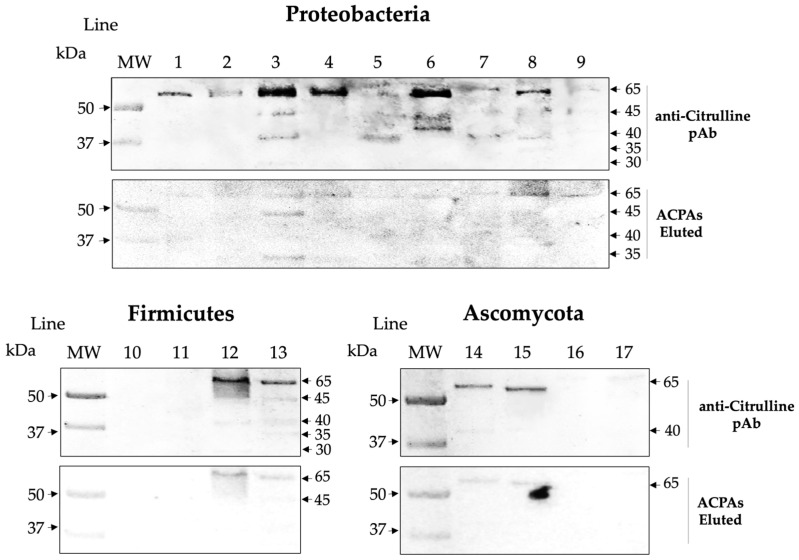
Coincident reactivity between eluted ACPAs against anti-citrulline pAb. The expression of citrullinated antigens is shown by reactivity with commercial anti-citrulline pAb and affinity ACPAs of RA patients. The lines correspond to proteins from different species: 1. *P. mirabilis*, 2. *P. vulgaris*, 3. *E. coli*, 4. *K. pneumoniae*, 5. *K. oxytoca*, 6. *C. freundii*, 7. *M. morgannii*, 8. *P. aeruginosa*, 9. *A. baumannii*, 10. *Lactobacillus* sp., 11. *E. faecalis*, 12. *S. epidermidis*, 13. *S. aureus*, 14. *C. albicans*, 15. *C. glabrata*, 16. *C. tropicalis*, and 17. *S. cerevisiae*.

**Figure 8 ijms-25-05192-f008:**
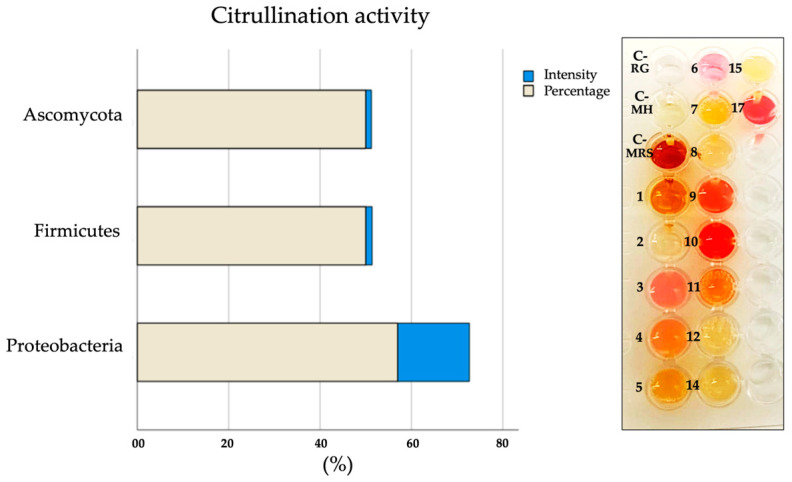
Determination of citrullination activity in microbial cultures. The percentage of species within each microbial phylum that presented citrullination activity is shown, as well as the amount of L-citrulline produced by the synthesis of nitric oxide (NO) represented as intensity (pinkish-reddish color). Griess reagent (RG), Mueller–Hinton agar (MH), and MRS agar. Firmicutes: 1. *Lactobacillus*, 2. *E. faecalis*, 3. *S. epidermidis*, and 4. *S. aureus*; Ascomycota: 5. *C. albicans*, 6. *C. glabrata*, 7. *C. tropicalis*, and 8. *S. cerevisiae*; and Proteobacteria: 9. *P. mirabilis*, 10. *P. vulgaris*, 11. *E. coli*, 12. *K. pneumoniae*, 13. *K. oxytoca (ND)*, 14. *C. freundii*, 15. *M. morgannii*, 16. *P. aeruginosa (ND)*, and 17. *A. baumannii*. ND = Not determined.

**Table 1 ijms-25-05192-t001:** Qualitative comparison of citrullination activity vs. expression of citrullinated antigens in the phyla Firmicutes, Proteobacteria, and Ascomycota.

Phyla	Firmicutes				Ascomycota				Proteobacteria								
Species	1	2	3	4	5	6	7	8	9	10	11	12	13	14	15	16	17
Citrullination activity	−	−	+	+	+	+	−	−	+	+	+	−	ND	−	−	ND	+
Expression of citrullinated antigens	−	−	+	+	+	+	−	−	+	+	+	+	−	+	+	+	+

Firmicutes: 1. *Lactobacillus*, 2. *E. faecalis*, 3. *S. epidermidis*, and 4. *S. aureus*; Ascomycota: 5. *C. albicans*, 6. *C. glabrata*, 7. *C. tropicalis*, and 8. *S. cerevisiae*; Proteobacteria: 9. *P. mirabilis*, 10. *P. vulgaris*, 11. *E. coli*, 12. *K. pneumoniae*, 13. *K. oxytoca* (ND), 14. *C. freundii*, 15. *M. morgannii*, 16. *P. aeruginosa* (ND), and 17. *A. baumannii*. ND = Not determined.

## Data Availability

No new data were created or analyzed in this study. Data sharing is not applicable to this article.

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
