# Peer review of "Protein Citrullination by Peptidyl Arginine Deiminase/Arginine Deiminase Homologs in Members of the Human Microbiota and Its Recognition by Anti-Citrullinated Protein Antibodies"

_ijms, 2024, doi:10.3390/ijms25105192_

Round 1
Reviewer 1 Report
Comments and Suggestions for Authors The paper by María-Elena Pérez-Pérez and co-workers is a cross-sectional study of cases and controls in patients with a diagnosis of RA and control subjects. This investigation was carried out to demonstrate the Peptidyl arginyl deiminases enzymes (PADs) and arginyl deiminases (ADs) homologs activity in cell extracts of various commensals such as members of the phyla Firmicutes, Proteobacteria and Ascomycota. The results demonstrated the presence of PADs/ADs homologs in 25 members of the phyla Firmicutes, Proteobacteria and Ascomycota, capable of endogenously deiminating arginine residues and producing citrullinated microbial proteins that are recognized by affinity ACPAs eluted of sera from RA patients. The paper is not convincing because the authors use only one technique and they do not perform other experiments to validate their scares results. The entire paper is based on immunoblotting analyses and the statistics over these data. It seems to me ridiculous that an appropriate antibody for loading control (or better more than one) has not been used. This is a common practice when you perform a western blotting to normalize all the obtained results. Moreover, in the Figure 2 the lanes of the first immunoblotting analysis are not correctly aligned e the number 9 is missing. Figure 5 lane 3, it is a real signal or is a deposit or a bubble? Please try to explain what happened to the sample. I do not like to read in the title the abbreviations as PADs and Ads because it doe not catch the audience, depriving the paper of potential readers.
Comments on the Quality of English Language
do not use abbreviations in the title.
Figure 2 is not well formatted
Author Response
Thank you very much for taking the time to review this manuscript. Please find the detailed responses below and the corresponding revisions/corrections highlighted/in track changes in the re-submitted files.

Reviewer 2 Report
Comments and Suggestions for Authors
Dear authors,
I recommend the publication of the manuscript after elucidate some details:
What is the significance of the human microbiome in relation to autoimmunity?
How do Peptidyl arginyl deiminases enzymes (PADs) contribute to rheumatoid arthritis (RA)?
What methods were employed in this study to investigate PADs and citrullinated proteins?
Which microbial phyla were found to harbor PADs/ADs homologs?
How do these enzymes impact the human microbiome, and what clinical implications might arise from their presence?
Author Response

(The authors gave the same response as above.)

Round 2
Reviewer 1 Report
Comments and Suggestions for Authors
In this revised version, the paper has been improved including one ausiliar technique. Unfortunately, one of my requests has not been understood properly. In particular, when I wrote It seems to me ridiculous that an appropriate antibody for loading control (or better more than one) has not been used... this means that I want to see the loading of each wb lane using an appropriate Ab (the author have chosen HSP70). So I do not understand why the authors show in Figure 1 and 2 the HSP70 signal relative to the 3 different species in toto and not the HSP70 signal in each lane. Moreover, looking at the panel of HSP70 in Figure 1 and 2, HSP70 also seems of different intensity/quantity in different species.
Please show me an immunoblot signal equal in each sample to be sure that you have loaded the same quantity of proteins in each lane.
Author Response

(The authors gave the same response as above.)

Round 3
Reviewer 1 Report
Comments and Suggestions for Authors
the paper is accepted in present form